



# What is the cause(s) of ozone trends in three megacity clusters in eastern China during 2015–2020?

Tingting Hu[1], Yu Lin[1], Run Liu[1,2], Yuepeng Xu[1], Boguang Wang[1,2], Yuanhang Zhang[3], Shaw Chen Liu[1,2]

[1]Institute for Environmental and Climate Research, Jinan University, Guangzhou, 511443, China

[2]Guangdong-Hongkong-Macau Joint Laboratory of Collaborative Innovation for Environmental Quality, Guangzhou, 511443, China

[3]State Key Joint Laboratory of Environmental Simulation and Pollution Control, College of Environmental Sciences and Engineering, Peking University, Beijing, 100871, China

*Correspondence to*: Run Liu (liurun@jnu.edu.cn), Shaw Chen Liu (shawliu@jnu.edu.cn)

**Abstract.** Thanks to a strong emission control policy, major air pollutants in China, including $PM_{2.5}$, $SO_2$, $NO_2$ and CO had shown remarkable reductions in 2015–2020. However, ozone ($O_3$) had increased significantly and emerged as a major air pollutant in eastern China at the same time. The annual mean concentration of maximum daily 8-hour average $O_3$ (MDA8) in three megacity clusters in eastern China, namely Beijing-Tianjin-Hebei (BTH), Yangtze River Delta (YRD) and Pearl River

Delta (PRD), showed alarming large upward linear trends of 2.4, 1.1 and 2.0 ppb yr$^{-1}$, respectively during the period 2015–2020. Furthermore, drastic trends of approximately three-fold increase in the number of $O_3$-exceeding days (defined as MDA8 $O_3$ >75 ppb) were observed during the same period. Our analysis of the upward trends of the annual mean concentration of MDA8 found that the trends were almost entirely attributable to the increase in the number of consecutive $O_3$-exceeding days. In addition, a widespread expansion of high $O_3$ from urban centers to surrounding rural regions was found in 2015–2017,

which had made the $O_3$ spatial distribution becoming more uniform after 2017. Finally, we found that the $O_3$ episodes with four or more consecutive $O_3$-exceeding days in the three megacity clusters were closely associated with the position and strength of the West Pacific subtropical high (WPSH), which contributed to the meteorological conditions characterized by clear sky, sinking motion and high vertical stability in the lower troposphere, and high solar radiation and positive temperature anomaly at the surface. These meteorological conditions were highly conducive to $O_3$ formation. Hence we hypothesize that

the cause of the worsening $O_3$ trends in BTH, YRD and PRD from 2015 to 2020 is attributable to the increased occurrence of meteorological conditions of high solar radiation and positive temperature anomaly under the influence of WPSH, tropical cyclones as well as mid-high latitude wave activities.

## 1 Introduction

Ozone ($O_3$) is an important greenhouse gas, which can also have adverse effects on human health, vegetation and materials

(Bell et al., 2006; Cohen et al., 2017; Kalabokas et al., 2020; Nuvolone et al., 2018). Surface $O_3$ is a secondary pollutant produced by photochemical reactions involving $O_3$ precursors such as volatile organic compounds (VOCs), carbon monoxide



(CO) and nitrogen oxides (NOx) (Ma et al., 2012; Monks et al., 2015; Wang et al., 2017). In addition to $O_3$ precursors, meteorological conditions are also important factors driving the $O_3$ formation. Solar radiation, temperature, relative humidity, wind speed, and cloud cover have been found to be closely related to $O_3$ formation (Yin et al., 2019; Dong et al., 2020; Han et

al., 2020). In addition, large-scale circulations, such as the East Asian monsoon, West Pacific subtropical high (WPSH) and tropical cyclones can influence $O_3$ concentration as well (Yang et al., 2014; Zhao and Wang, 2017; Lu et al., 2019; Rowlinson et al., 2019).

The concentrations of air pollutants $SO_2$, NOx, CO, $PM_{10}$ and $PM_{2.5}$ in China have been significantly reduced since 2013, thanks to the implementation of "Air Pollution Prevention and Control Action Plan". However, the $O_3$ concentration has

dramatically increased and emerged as a major air pollutant in eastern China (Zheng et al., 2018; Bian et al., 2019; Fu et al., 2019; Wang et al., 2020). $O_3$ concentrations are particularly high in the three megacity clusters in eastern China, namely Beijing-Tianjin-Hebei (BTH), Yangtze River Delta (YRD) and Pearl River Delta (PRD) (H. Liu et al., 2018; Guo et al., 2019; W. Yang et al., 2019; Gao et al., 2020; K. Li et al., 2021).

Annual mean concentrations of maximum daily 8-hour average (MDA8) $O_3$ in the three megacity clusters are shown in Figure

1. The linear increasing trends of MDA8 $O_3$ for BTH, YRD and PRD are 2.4, 1.1 and 2.0 ppb yr$^{-1}$, respectively during the period 2015–2020. These trends are unusually large compared to the trends in other parts of China as well as the trends worldwide (Lu et al., 2018; Chen et al., 2020; Zhang et al., 2020; Professional Committee of Ozone Pollution Control of Chinese Society for Environmental Sciences, 2022). Thus, an obvious scientific question is: What is the cause(s) of these large trends in $O_3$ concentration? Some recent studies suggested that enhanced photochemical processes induced by anthropogenic

emissions are responsible for these trends (Li et al., 2019; Wang et al., 2020; Shao et al., 2021). However, in our analysis of the $O_3$ trends at individual stations in eastern China during the period 2015–2020, we noticed that the interannual variations of $O_3$ concentration were strongly affected by the position and intensity of WPSH and the presence tropical cyclones in the western Pacific and South China Sea, consistent with the results of a number of recent studies (Zhao and Wang, 2017; Chang et al., 2019; Yin et al., 2019; Mao et al., 2020; Ouyang et al., 2022).These results suggest that transport/meteorological

parameters associated with WPSH and tropical cyclones may play an important role in the large trends of MDA8 $O_3$.

In this study, we focus on exploring possible contributions to the large $O_3$ trends in the three megacity clusters in eastern China by changes in meteorological parameters during the period 2015–2020. This paper is organized as follows. In Section 2, the data and methodology used in this study are described. Major characteristics of the $O_3$ trends in the three megacity clusters are discussed in Section 3.1. In Section 3.2, we examine the spatial expansion and saturation of high $O_3$. The annual change of $O_3$-

exceeding days with different durations are also examined. A hypothesis of the cause of $O_3$ trends in three megacity clusters in eastern China during 2015–2020 is presented in Section 3.3. Section 4 presents a summary and conclusions.





## 2 Data and methodology

### 2.1 Pollutant Data

In this study, the observed hourly concentrations of air pollutants, including $O_3$, $NO_2$, CO, $PM_{2.5}$, and $SO_2$ from 2015 to 2020
are obtained from the Chinese National Environmental Ministry of Environmental Protection (http://www.cnemc.cn/). Gridded
MDA8 $O_3$ data from Tracking Air Pollution in China (TAP) dataset (http://tapdata.org) with a resolution of 10 km are also
used (Xue et al., 2020).

### 2.2 Meteorological Data

The European Centre for Medium-Range Weather Forecasts (ECMWF) Reanalysis v5 (ERA5) dataset (available at
https://cds.climate.copernicus.eu/), with a horizontal resolution of $0.25° \times 0.25°$ and a time interval of 1 h, was used to analyze
the influence of climate change on $O_3$ pollution. The variables used in this study include 2m temperature (T2m), surface net
solar radiation (SSR), zonal and meridional wind at 500 hPa, and geopotential height at 500 hPa.

### 2.3 Methods

The Chinese National Ambient Air Quality Standard for MDA8 $O_3$ is 160 μg m$^{-3}$, which corresponds to 75 ppb at 273 K. It
follows that the $O_3$-exceeding days are defined as MDA8 $O_3$ concentration >75 ppb, while non-$O_3$-exceeding days are defined
as MDA8 $O_3$ concentration <75 ppb. According to the duration of $O_3$ pollution, it can be divided into consecutive $O_3$-exceeding
days with four or more days ($O_3$ days≥4) and consecutive $O_3$-exceeding days less than four days ($O_3$ days<4). In addition,
some common statistical methods are used in this study, including linear fitting, meteorological synthesis method, and two-
tailed Student's t test.
The normalized annual mean $O_3$ concentration of the $O_3$-exceeding days is calculated by adding the $O_3$ concentration of the
$O_3$-exceeding day each year and dividing it by the total number of days in the year. The normalized annual mean $O_3$ of the
non-$O_3$-exceeding days is calculated by the same method except for the non-$O_3$-exceeding days.
Table 1 lists the criteria and corresponding numbers of clean and polluted stations in the three megacity clusters. Clean and
polluted stations are defined basing on the frequency of $O_3$ pollution in 2015. Stations with the number of $O_3$-exceeding days
fewer than the clean criterion are considered as clean sites. When more than the polluted criterion, they are considered as
polluted sites. The criteria of the three regions are also listed in Table 1.

## 3 Results and discussion

### 3.1 Major characteristics of $O_3$ trends

Major characteristics of the large trends in the annual mean $O_3$ concentration are shown in Figures 2a, 2b and 2c for BTH,
YRD and PRD, respectively, in which the normalized annual mean concentrations of MDA8 $O_3$ in the three megacity clusters





are compared to contributions from two groups: The $O_3$-exceeding days and non-$O_3$-exceeding days. It is clear that the increase in $O_3$-exceeding days is the primary contributor to the large increase in the annual mean $O_3$ in all three megacity clusters from 2015 to 2020. Contributions from non-$O_3$-exceeding days are insignificant ($p > 0.1$), except that in BTH (Figure 2a) which shows a significant declining contribution ($p = 0.02$) due to the reduced number of non-$O_3$-exceeding days. Therefore, the following discussions on the $O_3$ trends will be focused on the $O_3$-exceeding days.

Annual numbers of single and consecutive $O_3$-exceeding days are shown in Figures 3a, 3b and 3c for BTH, YRD and PRD, respectively. A drastic two to three-fold increase in the annual numbers of consecutive $O_3$-exceeding days can be seen in all three regions. In contrast, the numbers of single $O_3$-exceeding days show only a small increase in PRD. These drastic increases in the annual numbers of consecutive $O_3$-exceeding days are clearly the primary contributors to the trends in $O_3$ shown in Figures 2a, 2b and 2c. This brings up some key scientific questions: What is the cause(s) of the drastic increases in the numbers of consecutive $O_3$-exceeding days? Is it due to changing $O_3$ photochemical processes or changing meteorological parameters? And will these drastic increases continue?

### 3.2 Spatial expansion and saturation of high $O_3$

Another important changing characteristic of $O_3$ concentrations is illustrated in Figure 4a, which depicts the annual mean concentrations of MDA8 $O_3$ in BTH during $O_3$-exceeding days for all 78 stations (black line), 14 stations in the highest category of $O_3$ concentration (average 103 ppb) observed in 2015 (red line, denoted polluted stations hereafter, Table 1) and 13 stations in the lowest category of $O_3$ (average 57 ppb) observed in 2015 (green line, denoted clean stations hereafter, Table 1). It is remarkable that $O_3$ concentrations at the clean stations caught up within 12 ppb with other stations in merely two years (an increase of about 30 ppb from 2015 to 2017), and actually equaled the average of other stations in 2019. Meanwhile, the polluted stations experienced a slight decrease in $O_3$ concentration, albeit not statistically significant. This phenomenon suggests strongly that the annual mean concentrations of MDA8 $O_3$ in BTH experienced a fast (within two years) and widespread spatial expansion of high $O_3$ from urban centers to surrounding regions where $O_3$ concentrations were low in 2015. Temporally most of the expansion was accomplished during 2015–2017. This phenomenon of a fast and widespread expansion of high $O_3$ concentrations from urban centers to surrounding regions were also observed at a slightly less degree in YRD (Figure 4b) and PRD (Figure 4c).

It is worth noting that $O_3$ concentrations at the polluted stations of about 100 ppb remained nearly constant throughout the entire period of 2015–2020, while the stations with $O_3$ concentration less than about 75 ppb in all three megacity clusters experienced a significant enhancement in $O_3$ concentration ($>5$ ppb yr$^{-1}$) during 2015–2017 (Figures 4a, 4b and 4c). This near-constant $O_3$ phenomenon suggests a saturation effect of $O_3$ formation when the annual mean concentration of MDA8 $O_3$ reached approximately 100 ppb.

The expansion and saturation of $O_3$ raises some interesting scientific questions: What is the cause(s) of the expansion and saturation? Why did the expansion and saturation happen mostly during 2015–2017? Did it have anything to do with the increase of consecutive $O_3$-exceeding days as suggested in Figure 3? These questions are addressed in the following section



by examining in detail the spatial expansion of high $O_3$ from urban centers to surrounding regions in BTH, YRD and PRD
during 2015–2017.

The spatial expansion of high $O_3$ from urban centers to surrounding regions in BTH and YRD during 2015–2017 can be clearly visualized in Figures 5 and 6, respectively. Figures 5a, 5b and 5c show the spatial distribution of daily mean concentrations of MDA8 $O_3$ for $O_3$-exceeding days in BTH in 2015, 2017 and their difference (2017 minus 2015), respectively. Comparing Figure 5a to 5b, one can see that the area inside the 80-ppb contour (75 ppb is the $O_3$ exceeding standard) expanded by about
a factor of five from 2015 to 2017. The daily average concentration of MDA8 $O_3$ within the BTH box increased from 66.42 ppb in 2015 (31 days, Fig. 5a) to 69.44 in 2017 (62 days, Fig. 5b), which was a difference of 3.02 ppb or a merely 4.5% increase between the two years (Fig. 5c). When accounted for the number of $O_3$-exceeding days, the ratio of MDA8 $O_3$ in all $O_3$-exceeding days between 2017 and 2015 became $(69.44 \times 62)/(66.42 \times 31) = 2.09$, implying that the increase in $O_3$ in BTH between 2015 and 2017 shown in Figure 2a (red line) was almost entirely (95.5%) due to the increase in the number of $O_3$-
exceeding days. These results together with those shown in Figure 3a suggest that the increase in $O_3$ in BTH between 2015 and 2017 was driven primarily by the increase of consecutive $O_3$-exceeding days. Spatially Figure 5c shows the expansion is mostly to the south and southwest outside of BTH, with YRD getting a lion's share of $O_3$ enhancements. Within the BTH box, the nearly constant concentrations of $O_3$ inside Beijing City (40 °N, 116.5 °E) coupled with the southwestward expansion of high $O_3$ in 2017 (Figure 5c) suggested that there was a saturation of $O_3$ inside Beijing City, and an expansion of weather
systems conducive to $O_3$ formation from Beijing toward the southwest of the BTH box during 2017 (Figures 5b and 5c). This change of weather conditions also caused significant increases in $O_3$ in YRD and even in southern China as far as the western PRD (Figure 5c). Nevertheless, the $O_3$ concentrations in YRD and PRD stayed below 70 ppb during the $O_3$-exceeding days of BTH in both 2015 (Figure 5a) and 2017 (Figure 5b). In other words, the $O_3$-exceeding days of YRD and PRD are mostly decoupled (i.e. not occurring at the same time) from those of BTH. A logical explanation for this phenomenon is that the
atmospheric conditions conducive to high $O_3$ formation in BTH do not overlap significantly with those conditions of YRD and PRD.

Figures 6a, 6b and 6c are the same as Figures 5a, 5b and 5c, respectively, except for YRD. Similar to BTH, one can clearly see the expansion of high $O_3$ from the vicinity of Shanghai City (31 °N, 121.3 °E) in the northwestern direction reaching as far as the central BTH box during the period 2015–2017 (Figure 6c). Comparing Figure 6a to 6b, one can see that the area inside
the 70-ppb contour expanded from Shanghai and vicinity northwestward by more than a factor of five from 2015 to 2017. This expansion was in different direction from the southwestern expansion occurred in BTH (Figure 5c). Since it is highly unlikely that any change in emissions could result in these different expansions in YRD and BTH, the logical explanation of the expansion in YRD would be that the weather system conducive to $O_3$ formation moved from the vicinity of Shanghai in 2015 (Figure 6a) northwestward toward western BTH in 2017 (Figures 6b and 6c). We note, however, this movement of the weather
system does not necessarily mean the direct transport of high $O_3$ or its precursors from the vicinity of Shanghai to northern BTH. In fact, the presence of separate rather than contiguous red patches of high $O_3$ (>70 ppb) in southern BTH and northern YRD in Figure 6b is a clear indication that the high $O_3$ are primarily controlled by local photochemical production from local





O₃ precursors under the expanded conducive weather conditions, rather than the direct upwind-downwind transport of high O₃ or its precursors. The daily average MDA8 O₃ in the YRD box increased from 53.79 ppb in 2015 (31 days, Fig. 6a) to 64.35

in 2017 (40 days, Fig. 6b), which was a difference of 10.6 ppb or a 20% increase between the two years (Fig. 6c). When accounted for the number of O₃-exceeding days, the ratio of MDA8 O₃ in all O₃-exceeding days between 2017 and 2015 became $(64.35 \times 40)/(53.79 \times 31) = 1.54$. This implied that the increase in O₃ in YRD between 2015 and 2017 shown in Figure 2b (red line) was due to both the increases in O₃ concentrations (+20%) and the number of O₃-exceeding days (+34%).

Figures 7a, 7b and 7c are the same as Figures 5a, 5b and 5c, respectively, except they are for PRD. Unlike BTH and YRD,

there was only a slight expansion of high O₃ within the PRD box toward the southwest in 2017 compared to 2015 (Figure 7c). Nevertheless, outside the PRD box there was an extensive expansion of high O₃ in eastern China, substantially greater the expansion within the PRD box (Figure 7c). The daily average concentration of MDA8 O₃ within the PRD box increased from 61.16 ppb in 2015 (14 days, Fig. 7a) to 65.18 in 2017 (36 days, Fig. 7b), which was a difference of 4.02 ppb or a merely 6.6% increase between the two years (Fig. 7c). After accounting for the number of O₃-exceeding days, the ratio of MDA8 O₃ in all

O₃-exceeding days between 2017 and 2015 became $(65.18 \times 36)/(61.16 \times 14) = 2.74$. This implied that the increase in O₃ in PRD between 2015 and 2017 shown in Figure 2c (red line) was almost entirely (93.4%) due to the increase in the number of O₃-exceeding days.

Figures 7a and 7b reconfirm that O₃-exceeding days in PRD are mostly decoupled from those in BTH (Figs. 5a and 5b) and YRD (Figs. 6a and 6b), as their spatial distributions are characterized by highly distinctive regional features in both 2015 and

2017. These differences suggest that the O₃-exceeding days mostly occur in different days in the three individual regions. On the other hand, a comparison of Figures 7c, 6c and 5c reveals a striking common feature of high values in southwestern BTH and northwestern YRD, and low values in eastern parts of all three BTH, YRD and PRD boxes. These common features suggest that the difference between 2015 and 2017 in all three individual regions is likely caused by a common mechanism/process that changed from 2015 to 2017. Moreover, as suggested in Figure 3, this common mechanism/process must be closely related

to higher number of consecutive O₃-exceeding days in 2017 over those of 2015.

More evidences against the emissions of air pollutants as a major cause of the expansion and saturation can be seen in Fig. 8, in which the annual mean concentrations of MDA8 O₃ during O₃-exceeding days are compared to those of Ox (O₃+NO₂) as well as other air pollutants in BTH in 2015–2020. The nearly 30 ppb increases in O₃ (Fig. 8a) at clean stations from 2015 to 2017 occurred also in Ox (Fig. 8b), suggesting that titration by NO or emission of NO was not the cause of the increases in O₃.

In addition, PM₂.₅ concentrations at polluted stations in Fig. 8c decreased significantly more than those at clean stations from 2015 to 2017, yet the clean stations experienced a near 30 ppb increase in O₃, while O₃ remained nearly constant at the polluted stations, suggesting that the proposed removal of HO₂ radicals by PM₂.₅ (Li et al., 2020; Shao et al., 2021) was also not a likely cause of the increases in O₃. Finally, neither CO nor NO₂ showed any significant change at clean stations between 2015 and 2017, implying negligible change in O₃ precursors, NOx and VOCs, as their emission rates tended to be proportional to those

of CO and NO₂. This again supported the notion that changes in the emissions of O₃ precursors were unlikely to be the cause of the increases in O₃ at clean stations from 2015 to 2017.





## 3.3 Cause(s) of the expansion and saturation

Major findings of subsections 3.1 and 3.2 can be summarized as follows: (1) The trends in $O_3$ observed in the three megacity clusters in eastern China during 2015–2020 (Figure 1) were mainly caused by the large trends of approximately two to three-fold increase in the number of consecutive $O_3$-exceeding days (Figures 2 and 3). (2) A fast and widespread expansion of high $O_3$ from urban centers to surrounding regions was observed in the three megacity clusters during 2015–2019 (Figure 4); and the majority of the expansions were accomplished during the 2015–2017 period (green lines in Figure 4). (3) The expansions of high $O_3$ in the three megacity clusters were accompanied by a saturation effect that $O_3$ concentrations at the polluted stations (high $O_3$ in 2015) of about 100 ppb remained nearly constant throughout the entire period of 2015–2020, while the clean stations (low $O_3$ in 2015) with $O_3$ of about 75 ppb in all three megacity clusters in 2015 experienced a significant enhancement in $O_3$ (>5 ppb yr$^{-1}$) during 2015–2017 (Figures 4a, 4b and 4c). And (4) There are independent evidences, including spatial distribution of the expansion (Figs. 5 and 6) and inter-annual variations in $O_3$, Ox, $NO_2$, CO and $PM_{2.5}$ (Figure 8), suggest that transport/meteorology rather than emissions of $O_3$ precursors is more likely to be the major cause of the expansion and saturation.

While a specific process/mechanism has yet to be found as the primary contributor to the trends in $O_3$ observed in the three megacity clusters, the findings summarized above suggest that an examination into transport/meteorological processes involved in $O_3$ episodes with consecutive $O_3$-exceeding days could provide useful information on the identity of the primary contributor. Using BTH as an example, we address this issue in the following by dividing $O_3$ episodes of a given year into two groups: the first group has four or more consecutive $O_3$-exceeding days (labeled $O_3$ days≥4), the other has less than four consecutive $O_3$-exceeding days (labeled $O_3$ days<4). Figure 9a shows the mean daily $O_3$ concentrations of the first group in 2015 (mean concentration of 71.14 ppb inside the BTH box, 7 days), Figure 9b shows the mean daily $O_3$ concentrations of the second group (65.04 ppb, 24 days), and Figure 9c is the difference between the two groups (6.1 ppb, Table 2). Figures 9d–9f are the same as Figures 9a–9c, respectively, except for 2017. The first group in 2017 had 28 days and mean $O_3$ of 74.43 ppb inside the BTH box, while the second group had 34 days and 65.32 ppb (Table 2). One of most remarkable differences between 2017 and 2015 in Figures 9a–9f was the large number of days with four or more consecutive $O_3$-exceeding days (first group) in 2017 (28 days, Figure 9d) over that of 2015 (7 days, Figure 9a), which alone contributed to about 62% of the difference in $O_3$ between 2017 and 2015 as shown in Fig. 2a (red line). Approximately 30% was contributed by the 10 days' difference (2017 vs. 2015) in the number of days with less than four consecutive $O_3$-exceeding days (second group). The contribution by the higher average concentration of MDA8 $O_3$ of the first group in 2017 is only about 8% (Table 2). These values of contributions reconfirm what is shown in Figure 3a, i.e., the greater frequency of episodes with four or more consecutive $O_3$-exceeding days contributes the majority (62%) to the higher $O_3$ in BTH in 2017 vs. 2015, the greater intensity/concentration of $O_3$ during the episodes contributes only about 8%, consistent with the expansion and saturation effect discussed earlier. The phenomenon of frequency over intensity is even more pronounced when the data of 2015 (4$^{th}$ row and 4$^{th}$ column in Table 2)





are compared to those of 2019 (8[th] row and 4[th] column in Table 2), in which the higher frequency of the first group of 2019

contributes as much as 83% to the higher $O_3$ in BTH in 2019 vs. 2015.

The phenomena illustrated in Figures 9a–9f also exist in YRD and PRD as well as in most other years. Figures equivalent to Figures 9a–9c for all years in the three city clusters (except PRD during 2015–2016, in which no episode with four or more consecutive $O_3$-exceeding days occurred) are provided in the Supplementary Material (Figures S1, S2 and S3). Essential information derived from those figures is summarized in Tables 2–4. The 4[th] column of Table 2 shows that the number of days

with four or more consecutive $O_3$-exceeding days in BTH increased consistently from 7 days in 2015 to 66 days 2019, and dropped back to 38 days in 2020; this pattern of changes matched very well with those in Figure 2a (red line). The same can be said for YRD (Table 3) and PRD (Table 4), except there are some minor contributions from the 3[rd] column in Tables 3 and 4, i.e., days with less than four consecutive $O_3$-exceeding days. Another remarkable point is that the difference between (>=4days) and (<4days) (5th column) in Tables 2–4 is slightly positive (mostly by a few percent) for all three city clusters in

all years, which again implies expansion and saturation of high $O_3$ in episodes with four or more consecutive $O_3$-exceeding days. In summary, Tables 2–4 show quantitatively that the temporal and spatial changes in $O_3$ concentrations in three megacity clusters of eastern China during 2015–2020 can be mostly attributed to the changes in the number of days with four or more consecutive $O_3$-exceeding days. It follows then that the critical question of our quest for the cause(s) of the remarkable large upward linear trend in $O_3$ of the three megacity clusters becomes: what process/mechanism is conducive to the formation of

$O_3$ episodes with four or more consecutive $O_3$-exceeding days?

Mao et al. (2020) made a comprehensive study of an 11-day $O_3$ episode in BTH in 2017 and found it was dominated by the presence of the WPSH and mid-high latitude wave activities. Depending on the position and intensity, WPSH is well known to be an important factor affecting $O_3$ concentrations in various parts of eastern China (Zhao and Wang, 2017; Chang et al., 2019; Yin et al., 2019). During this 11-day $O_3$ episode, the ridge line of WPSH maintained at approximately 22 °N from June

24 to June 29, which in combination with mid-high latitude wave activities induced meteorological conditions highly conducive to the $O_3$ production in BTH and northern YRD (Mao et al., 2020).

Ouyang et al. (2022) made a modeling study of an 11-day $O_3$ episode (09/22–10/02) in PRD in 2019. They found that during this $O_3$ episode PRD was mainly influenced by the WPSH followed by Typhoon Mina. From September 25 to 28, the area enclosed by the 5880 gpm isoline of the 500 hPa geopotential height of WPSH continued to cover the entire PRD, and the

downdraft associated with WPSH suppressed the vertical dispersion of air pollutants, and the intense solar radiation under clear sky conditions (caused by the downdraft) were highly conducive to the photochemical production of $O_3$. Afterward Typhoon Mina moved northward across the ridge of WPSH to the southeast of Taiwan, and PRD region was under the downdraft in the periphery of Mina until it passed over Korea on October 3, 2019. Even then the influence of WPSH on $O_3$ formation remained strong as evident by the fact that PRD stayed near or above the 5860 gpm isoline (Ouyang et al., 2022)

when Typhoon Mina moved from east of the Philippines (September 29) to South Korea (October 3).

These two recent studies demonstrated the important role played by WPSH on the $O_3$ episodes with consecutive $O_3$-exceeding days. Following the method used by Mao et al. (2020), Figure 10 depicts the composite 500 hPa geopotential heights, humidity



and winds for all four $O_3$ episodes with four or more consecutive $O_3$-exceeding days in BTH in 2017, namely 05/16–05/21 (Figure 10a), 06/14–06/21 (Figure 10b), 06/25–07/03 (Figure 10c) and 07/10–07/14 (Figure 10d). The case of Figure 10c is

identical to the case investigated by Mao et al. (2020). And Figure 10c is highly consistent with their corresponding figures for 06/30 and 07/01, which were characterized by a remarkable belt of high humidity north and west of the 5880 gpm isoline of WPSH. The high humidity contributed to extensive clouds and precipitation and thus low $O_3$ formation over southern YRD and southern China. North to the belt there was a large patch of low humidity over northern China under the control of anticyclonic flow, which contributed to the meteorological conditions characterized by clear sky, sinking motion and high

vertical stability in the lower troposphere, as well as high SSR and positive T2m anomaly at the surface in BTH and northern YRD. These meteorological conditions were highly conducive to the $O_3$ formation in BTH and northern YRD. Figures 10a, 10b and 10d share the same essential characteristics of Figure 10c, including the belt of high humidity north and west of the 5880 gpm isoline of WPSH, as well as the large patch of low humidity over northern China under the control of anticyclonic flows.

In Figures 11a and 11b the values of SSR and T2m of the episodes with four or more consecutive $O_3$-exceeding days are compared to those of $O_3$ episodes with less than four consecutive $O_3$-exceeding days, and to those of clean days (non-$O_3$-exceeding days). As expected, the $O_3$ episodes with four or more consecutive $O_3$-exceeding days consistently have the highest values of SSR and T2m, while the clean days have the lowest values. This is the case in nearly all years studied as shown in the Supplementary Material (Figure S4), and is also generally true in YRD and PRD (Figures S5 and S6). Coupling the higher

values of SSR and T2m in the $O_3$ episodes with four or more consecutive $O_3$-exceeding days depicted in Figure 11 and greater number of days in the $O_3$ episodes with four or more consecutive $O_3$-exceeding days shown in Figure 3, we therefore propose a hypothesis as follows: the cause of worsening $O_3$ trends in BTH, YRD and PRD from 2015 to 2020 could be attributed to the increased occurrence of meteorological conditions of high solar radiation and positive temperature anomaly at the surface under the influence of WPSH, tropical cyclones as well as mid-high latitude wave activities.

Quantitatively the coupling of Figure 11 with Figure 3 can be performed by multiplying the difference between the red (four or more consecutive $O_3$-exceeding days) and green (clean days) values of SSR/T2m in Figure 11 with the frequency of occurrence (in percentage of total days) of $O_3$ episodes with four or more consecutive $O_3$-exceeding days from Figure 3. The results are compared to the yearly total $O_3$-exceeding days in Figure 12. Agreement between the yearly $O_3$-exceeding days and weighted SSR is very good with R values 0.88 or greater in all three regions, lending strong support for our hypothesis.

Agreement between the yearly $O_3$-exceeding days and weighted T2m is good in BTH but poor in YRD and PRD, which probably suggests that T2m is not as strongly coupled to $O_3$ formation as SSR. Inclusion of $O_3$ episodes with less than four consecutive $O_3$-exceeding days in Figure 12 did not change the correction coefficients significantly, supporting the robustness of results shown in the figure.

Figure 12 brings the closure to our quest for the cause(s) of the remarkable large upward linear trends in $O_3$ of the three

megacity clusters as follows: The cause is the increased occurrence of meteorological conditions associated with four or more consecutive $O_3$-exceeding days which are characterized by clear sky, sinking motion and high vertical stability in the lower



troposphere, and high solar radiation and positive temperature anomaly at the surface, which are highly conducive to $O_3$ formation. These meteorological conditions are driven primarily by certain positions and strengths of WPSH depending on the region, which are often exacerbated by the presence of tropical cyclones (Ouyang et al., 2022) as well as mid-high latitude
wave activities (Mao et al., 2020).

## 4 Summary and Conclusions

Thanks to a strong emission control policy, major air pollutants in China, including $PM_{2.5}$, $SO_2$, $NO_2$ and CO had shown remarkable reductions during 2015–2020. However, $O_3$ concentration had increased significantly and emerged as a major air pollutant in eastern China during the same time period. The annual mean concentration of MDA8 in three megacity clusters
in eastern China, namely BTH, YRD and PRD, showed alarming large upward linear trends of 25%, 10% and 19%, respectively during 2015–2019. Identifying the causes of these worsening $O_3$ trends is urgently required for air pollution prevention and management.

Some recent studies suggested that enhanced photochemical processes induced by changing anthropogenic emissions were responsible for these trends (Li et al., 2019; Wang et al., 2020; Shao et a., 2021). However, we noticed that there were
independent evidences, including the spatial distribution of the expansion of high $O_3$ (Figures 5 and 6) and inter-annual variations in $O_3$, Ox, $NO_2$, CO and $PM_{2.5}$ (Figure 8), suggesting that transport/meteorological conditions rather than emissions of $O_3$ precursors were more likely to be the major contributor to the $O_3$ trends. Moreover, we found that the trends in $O_3$ observed in the three megacity clusters during 2015–2020 (Figure 1) were mainly caused by the large trend of approximately two to three-fold increase in the number of consecutive $O_3$-exceeding days (Figure 3), during that time a fast and widespread
expansion of high $O_3$ from urban centers to surrounding regions was observed (Figure 4), and the majority of the expansions was accomplished during the two-year 2015–2017 period (green lines in Figure 4). Furthermore, the expansions of high $O_3$ in the three megacity clusters were accompanied by a saturation effect that $O_3$ concentrations at the polluted stations (high $O_3$ in 2015) of about 100 ppb remained nearly constant throughout the entire period of 2015–2020, while the clean stations (low $O_3$ in 2015) with $O_3$ less than 75 ppb in all three megacity clusters experienced a significant enhancement in $O_3$ (>5 ppb yr$^{-1}$)
during 2015–2017 (Figures 4a, 4b and 4c). Finally, greater frequency of episodes with four or more consecutive $O_3$-exceeding days contributed the majority to the higher $O_3$ in all three megacity clusters in 2017 vs. 2015, the greater intensity/concentration of $O_3$ during the episodes contributes only about 10% (Figure 9), consistent with the expansion and saturation effect discussed earlier.

Two recent studies (Mao et al., 2020; Ouyang et al., 2022) demonstrated the important role played by WPSH on the $O_3$ episodes
with consecutive $O_3$-exceeding days. Following the method used by Mao et al. (2020), we show in Figure 10 the composite 500 hPa geopotential heights, humidity and winds for each of the $O_3$ episodes with four or more consecutive $O_3$-exceeding days in BTH in 2017. To the north and west of the 5880 gpm isoline of WPSH, there existed a large patch of low humidity over northern China under the control of anticyclonic flow, which contributed to the meteorological conditions characterized





by clear sky, sinking motion and high vertical stability in the lower troposphere, and high solar radiation and positive
temperature anomaly at the surface in BTH and northern YRD (Figure 10). These meteorological conditions were highly
conducive to the $O_3$ formation in BTH and northern YRD. The meteorological conditions of high solar radiation and positive
temperature anomaly at the surface are particularly significant in the episodes with four or more consecutive $O_3$-exceeding
days in BTH (Figures 11a and 11b). This is the case in nearly all years studied and is also generally true for YRD and PRD.
Coupling the higher values of SSR and T2m in the $O_3$ episodes with four or more consecutive $O_3$-exceeding days depicted in
Figure 11 and greater occurrence (number of days) in the $O_3$ episodes with four or more consecutive $O_3$-exceeding days shown
in Figure 3, we hypothesize that the cause of the worsening $O_3$ trends in BTH, YRD and PRD from 2015 to 2020 could be
attributed to the increased occurrence of meteorological conditions of high solar radiation and positive temperature anomaly
under the influence of WPSH, tropical cyclones as well as mid-high latitude wave activities. The hypothesis is substantiated
in Figure 12, which shows excellent agreement between the yearly $O_3$-exceeding days and SSR with R values 0.88 or greater
in all three regions. Agreement between the yearly $O_3$-exceeding days and T2m is good in BTH but poor in YRD and PRD,
which probably suggests that T2m is not as strongly coupled to $O_3$ formation as SSR. Figure 12 brings the closure to our quest
for the cause(s) of the remarkable large upward linear trends in $O_3$ of the three megacity clusters as follows: The cause is the
increased occurrence of meteorological conditions characterized by clear sky, sinking motion and high vertical stability in the
lower troposphere, and high solar radiation and positive temperature anomaly at the surface.
In conclusion, we believe that changes in the meteorological conditions are the main reason for the dramatic aggravation of
$O_3$ pollution in the three megacity clusters in eastern China in 2015–2020. Therefore, we suggest that future $O_3$ pollution
prevention and control policies should pay more attention to changes in the meteorological/climate conditions, particularly
changes in the large-scale circulations, including WPSH, tropical cyclones and the Asian summer monsoon.

*Data availability.* Hourly surface $O_3$, $PM_{2.5}$, CO and $NO_2$ data were obtained from China National Environmental Centre
(http://www.cnemc.cn/en/, last access: 10 July 2022). Hourly meteorological data are obtained from European Centre for
Medium-Range Weather Forecasts ERA5 reanalysis (https://www.ecmwf.int/, last access: 10 July 2022). The data of this paper
are available upon request to Shaw Chen Liu (shawliu@jnu.edu.cn).

*Author Contributions.* SL and RL proposed the essential research idea. TH, and YL performed the analysis. TH, YL, RL, and
SL drafted the manuscript. YX, BW, and YZ helped analysis and offered valuable comments. All authors have read and agreed
to the published version of the manuscript.

*Competing interests.* The authors declare that they have no conflict of interest.


*Acknowledgments.* The authors thank the China National Environmental Centre and European Centre for Medium-Range
Weather Forecasts for providing datasets that made this work possible. We also acknowledge the support of the Institute for





Environmental and Climate Research and Guangdong-Hongkong-Macau Joint Laboratory of Collaborative Innovation for Environmental Quality in Jinan University.


*Financial support.* This research was supported by the National Natural Science Foundation of China (grant number 92044302, 41805115), Guangzhou Municipal Science and Technology Project, China (grant number 202002020065), Special Fund Project for Science and Technology Innovation Strategy of Guangdong Province (grant number 2019B121205004), Guangdong Innovative and Entrepreneurial Research Team Program (grant number 2016ZT06N263), and National Key

Research and Development Program of China (grant number 2018YFC0213906).



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



**Table 1.** Criteria and corresponding numbers of clean and polluted stations in the three megacity clusters in 2015.

|  | Criterion of clean sites | Number of clean sites | Criterion of polluted sites | Number of polluted sites | Total number of sites |
|---|---|---|---|---|---|
| BTH | ≤19 days | 13 | ≥ 71 days | 14 | 78 |
| YRD | ≤ 37 days | 54 | ≥ 67 days | 13 | 152 |
| PRD | ≤ 12days | 10 | ≥46 days | 10 | 48 |



**Table 2.** Mean $O_3$ concentrations (ppb) and number of days of all $O_3$-exceeding days (2nd column), consecutive $O_3$-exceeding days with less than four days (3rd column), consecutive $O_3$-exceeding days with four or more days (4th column) and the difference between (≥4days) and (<4days) (5th column) within the BTH box in 2015–2020.

| | All days Concentration (days) ppb | <4 days Concentration (days) ppb | ≥4 days Concentration (days) ppb | Difference (≥4 days) – (<4 days) ppb |
|---|---|---|---|---|
| 2015 | 66.42(31) | 65.04(24) | 71.14(07) | 6.10 |
| 2016 | 64.13(43) | 62.65(26) | 66.39(17) | 3.74 |
| 2017 | 69.44(62) | 65.32(34) | 74.43(28) | 9.11 |
| 2018 | 68.21(74) | 65.43(27) | 69.80(47) | 4.37 |
| 2019 | 70.19(96) | 65.28(30) | 72.42(66) | 7.14 |
| 2020 | 69.69(78) | 65.52(40) | 74.08(38) | 8.56 |



**Table 3.** Same as Table 2, but for YRD.

|  | All days Concentration (days) ppb | <4 days Concentration (days) ppb | ≥4 days Concentration (days) ppb | Difference (≥4 days) – (<4 days) ppb |
|---|---|---|---|---|
| 2015 | 53.79(31) | 53.59(19) | 54.11(12) | 0.52 |
| 2016 | 58.87(27) | 58.03(23) | 63.73(04) | 5.70 |
| 2017 | 64.35(40) | 62.62(25) | 67.22(15) | 4.60 |
| 2018 | 63.33(43) | 62.49(32) | 65.75(11) | 3.26 |
| 2019 | 67.18(49) | 66.09(27) | 68.51(22) | 2.42 |
| 2020 | 65.84(38) | 64.12(27) | 70.06(11) | 5.94 |



**Table 4.** Same as Table 2, but for PRD.

|  | All days<br>Concentration (days)<br>ppb | <4 days<br>Concentration (days)<br>ppb | ≥4 days<br>Concentration (days)<br>ppb | Difference<br>(≥4 days) – (<4 days)<br>ppb |
|---|---|---|---|---|
| 2015 | 61.16(14) | 61.16(14) | ---(0) | --- |
| 2016 | 58.44(19) | 58.44(19) | ---(0) | --- |
| 2017 | 65.18(36) | 64.60(23) | 66.20(13) | 1.60 |
| 2018 | 65.82(31) | 63.27(16) | 68.55(15) | 5.28 |
| 2019 | 69.80(62) | 65.96(29) | 73.16(33) | 7.20 |
| 2020 | 65.08(37) | 63.87(22) | 66.84(15) | 2.97 |





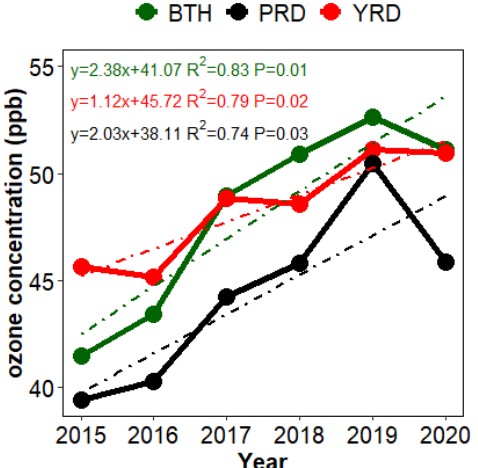

**Figure 1: Annual mean concentrations of maximum daily 8-hour average O₃ (MDA8) in BTH (green), YRD (red) and PRD (black).**





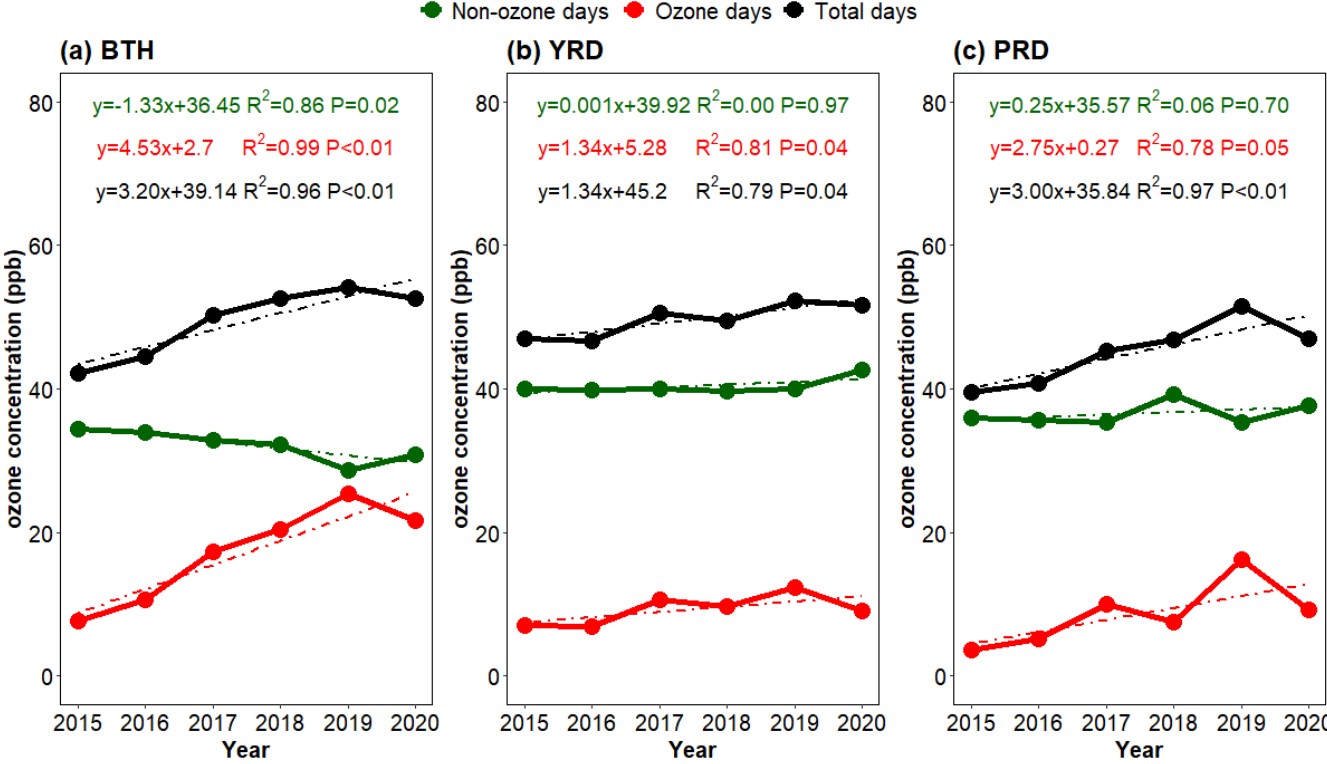

**Figure 2: Contributions from the O₃-exceeding days (red) and non-O₃-exceeding days (green) to the annual mean concentration of maximum daily 8-hour average O₃ (black) in BTH (a), YRD (b) and PRD (c).**





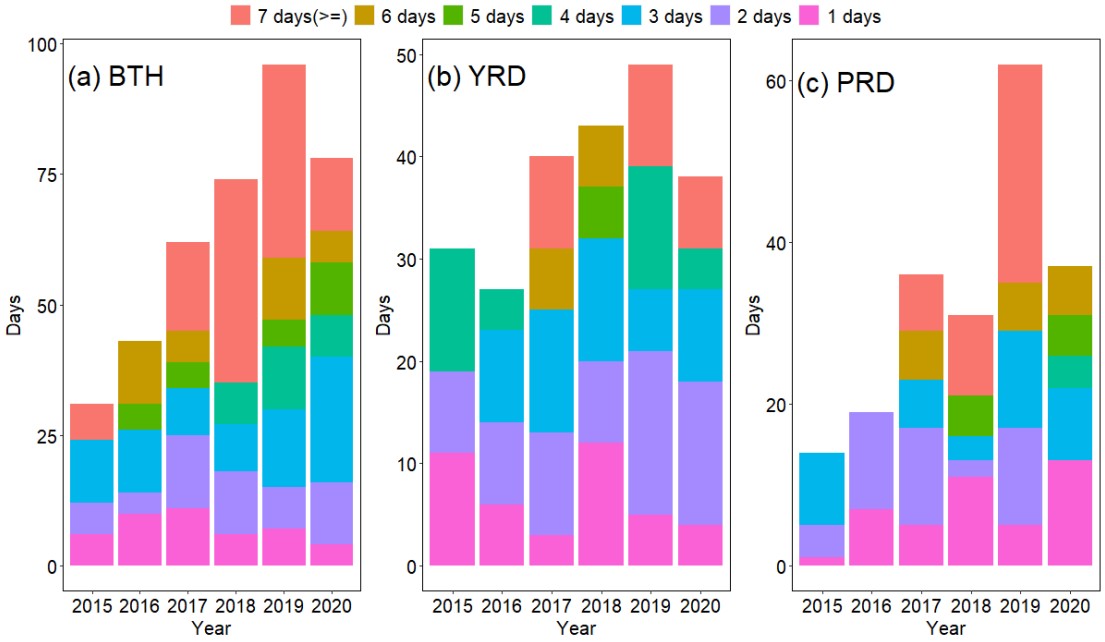

**Figure 3: Annual numbers of various consecutive O₃-exceeding days in BTH (a), YRD (b) and PRD (c). Individual colors denote different numbers of consecutive O₃-exceeding days.**



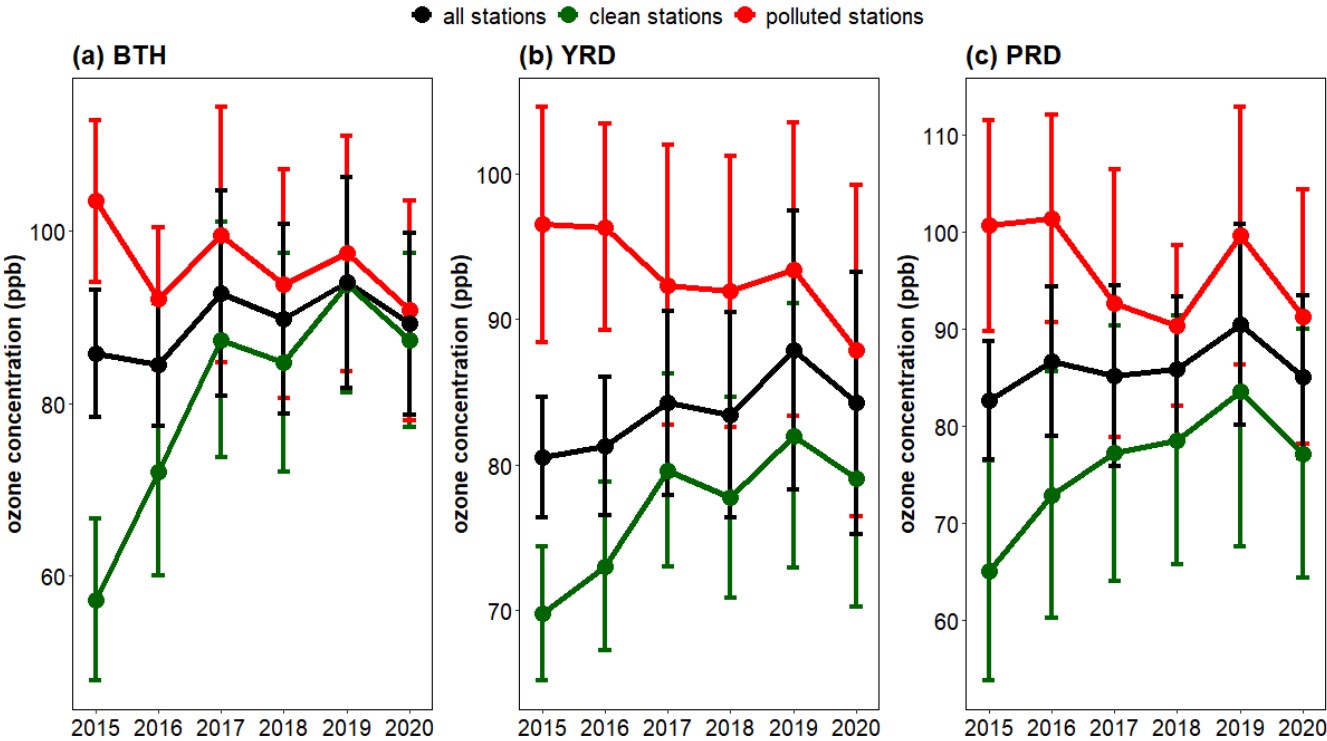

**Figure 4: Mean concentrations of maximum daily 8-hour average O₃ during O₃-exceeding days for all stations (black), polluted stations (red) and clean stations (green) in BTH (a), YRD (b) and PRD (c).**


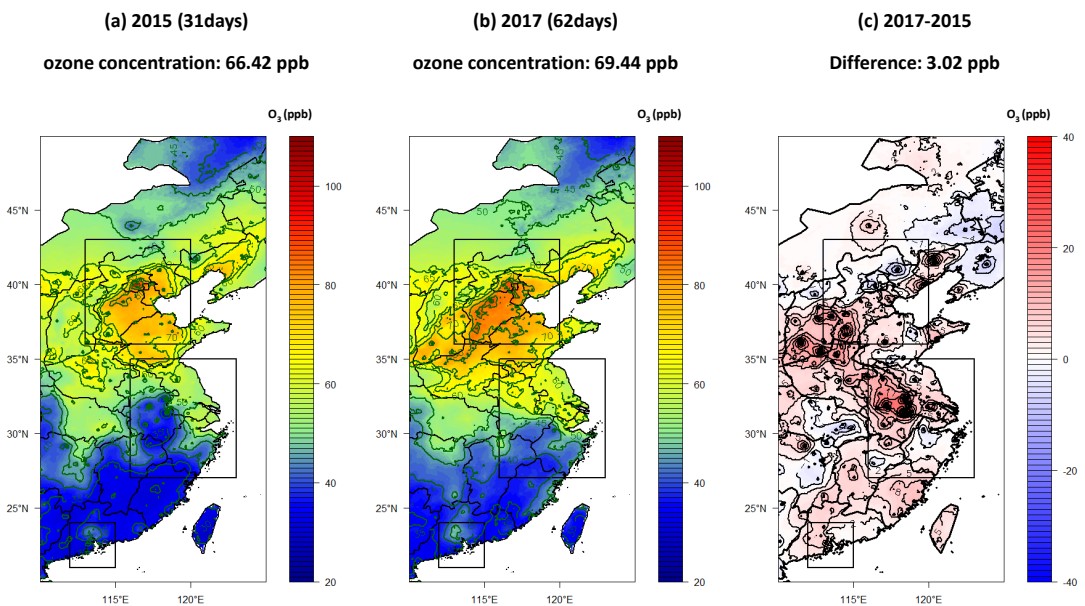


**Figure 5: Spatial distribution of annual mean concentrations of maximum daily 8-hour average O₃ for O₃-exceeding days in BTH in 2015 (a), 2017 (b) and their difference (2017 - 2015) (c). The top, middle and bottom rectangle boxes denote BTH, YRD and PRD districts, respectively. The number inside the parenthesis behind 2015 or 2017 denotes the number of O₃-exceeding days.**


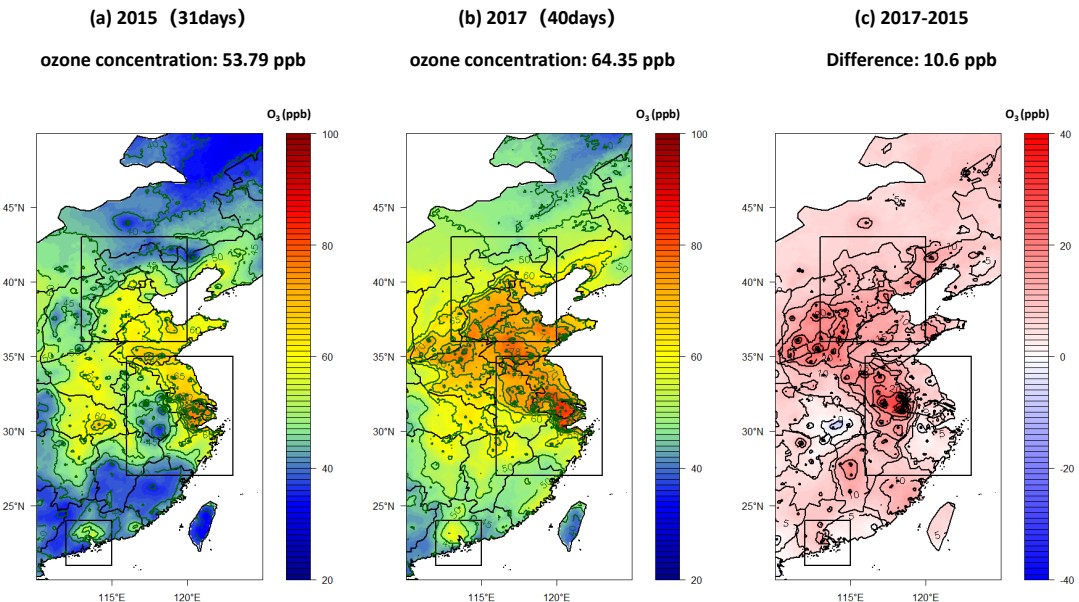

Figure 6: Same as Figure 5 except for YRD.





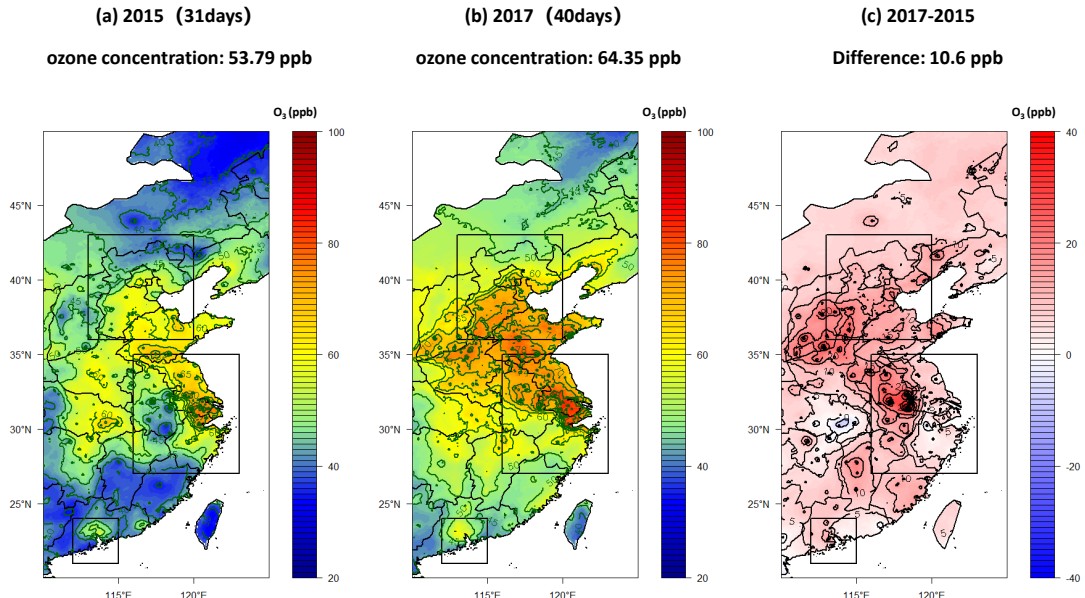

**Figure 7: Same as Figure 5 except for PRD.**



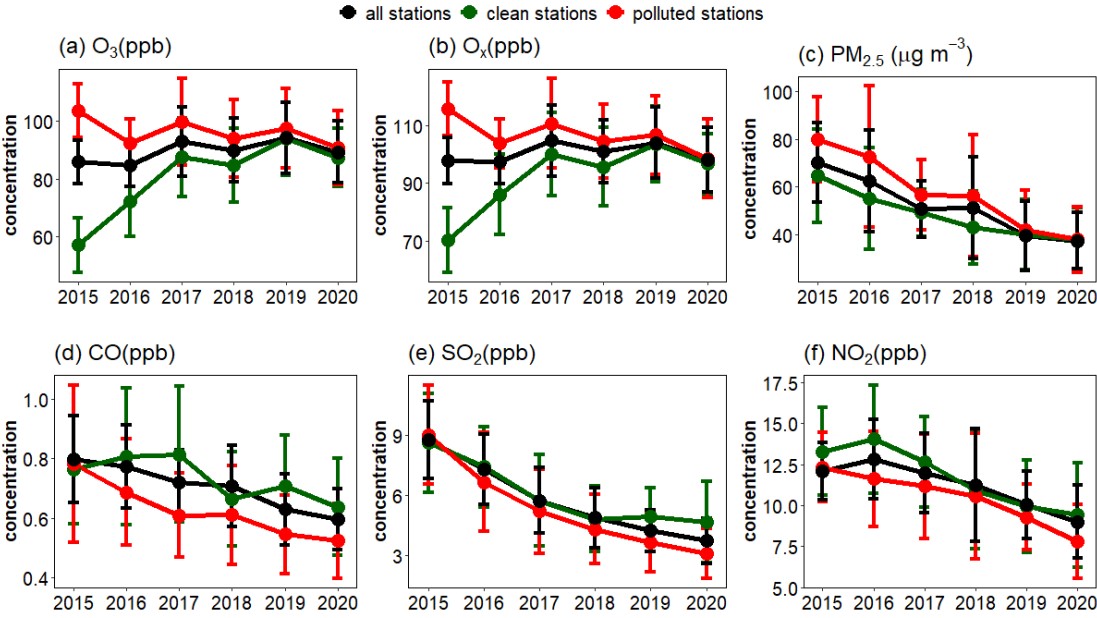


**Figure 8: Annual mean concentrations of maximum daily 8-hour average O₃ in BTH during O₃-exceeding days for all stations (black), polluted stations (red) and clean stations (green) (a), same as (a) except for Ox (b), PM₂.₅ (c), CO (d), SO₂ (e), NO₂ (f).**





**(a) 2015: O$_3$ days>=4 (7days)**

**ozone concentration: 71.14 ppb**

**(b) 2015: O$_3$ days<4 (24days)**

**ozone concentration: 65.04 ppb**

**(c) Difference**

**ozone concentration: 6.1 ppb**

**(d) 2017: O$_3$ days>=4 (28days)**

**ozone concentration: 74.43 ppb**

**(e) 2017: O$_3$ days<4 (34days)**

**ozone concentration: 65.32 ppb**

**(f) Difference**

**ozone concentration: 9.11 ppb**

**Figure 9: Spatial distribution of daily mean MDA8 O$_3$ of O$_3$-exceeding days in BTH for O$_3$ episodes with four or more consecutive**
**O$_3$-exceeding days in 2015 (a), O$_3$ episodes with less than four consecutive O$_3$-exceeding days in 2015 (b), and (b minus a) (c); (d, e and f) are the same as (a, b and c), respectively, except for 2017.**



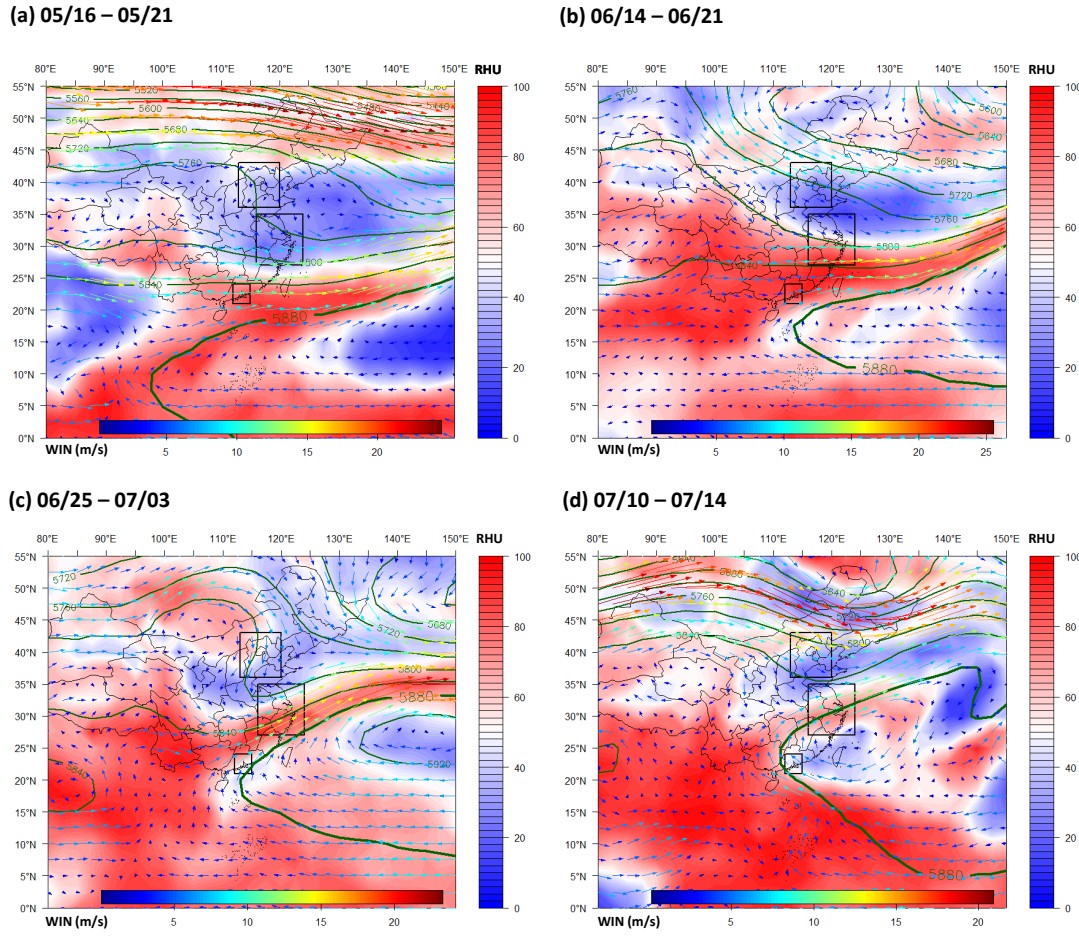

**Figure 10: Composite 500 hPa geopotential height contours, humidity and winds for four O₃-exceeding episodes in BTH in 2017: 05/16–05/21 (a), 06/14–06/21 (b), 06/25–07/03 (c) and 07/10–07/14 (d).**




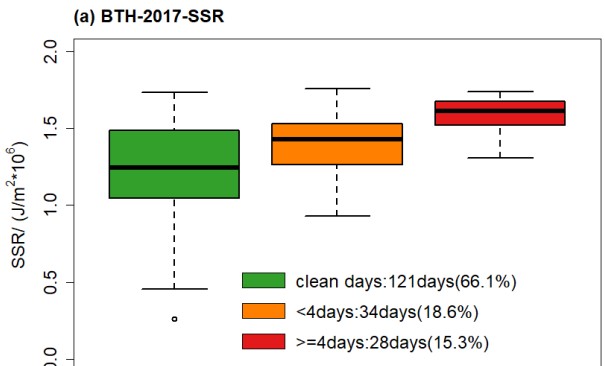

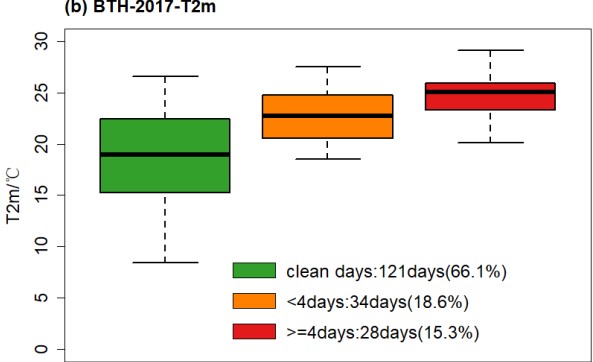

**Figure 11: Surface solar radiation (SSR) (a) and temperature (T2m) (b) in BTH in 2017 for four episodes with four or more consecutive O$_3$-exceeding days (red), clean days (non-ozone-exceeding days) (green) and ozone episodes with less than four consecutive O$_3$-exceeding days (orange).**





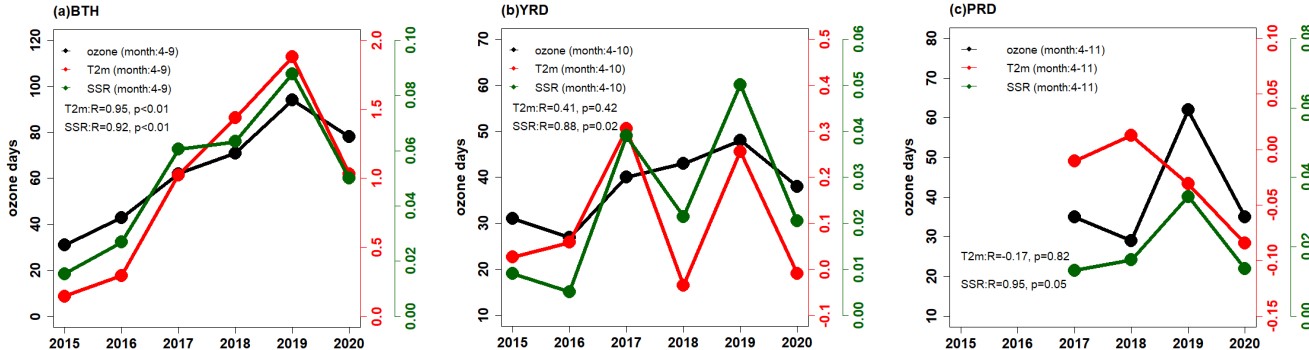


**Figure 12. Correlations among annual O₃-exceeding days, surface solar radiation (SSR) and temperature (T2m) in BTH (a), YRD (b) and PRD (c).**