# Peer review of "What is the cause(s) of ozone trends in three megacity clusters in eastern China during 2015–2020?"

_Atmospheric Chemistry and Physics, 2022_

## Author Comment (AC1)

**Erratum: What is the cause(s) of ozone trends in three megacity clusters in eastern China during 2015–2020?**

This note aims to correct an error in Figure 7 of Hu et al. (2023). In the version of this article initially published, Figure 7 was erroneously pasted. The interpretation of the results remains unchanged.

[Figure]

Figure 7. Spatial distribution of mean concentrations of maximum daily 8 hour average O$_3$ for ozone-exceeding days in PRD in (a) 2015, (b) 2017 and (c) their difference (2017-2015). The top, middle and bottom rectangle boxes denote BTH, YRD and PRD districts, respectively.

---

## Author Comment (AC2)

Dear Editor,

We appreciate the prompt reviews and would like to thank the reviewers for insightful comments and suggestions on our manuscript entitled "What is the cause(s) of ozone trends in three megacity clusters in eastern China during 2015–2020?" (MS No.: acp-2022-781). We have carefully considered all comments and suggestions. Listed below are our point-by-point responses to all comments and suggestions of Reviewer #1 (Reviewer's points in black, our responses in blue).

**Referee #1**

The annual mean concentration of MDA8 increased at a high rate in BTH, YRD and PRD during the period 2015-2020. The spatial expansion of high $O_3$ from urban centers to surrounding regions was found during 2015-2017, accompanied by a saturation effect. The authors suggest that the occurrence of meteorological conditions of solar radiation and positive temperature under WPSH and mid-high latitude wave activities are the main reason for the increased $O_3$ episodes with four or more consecutive $O_3$-exceeding days in the three megacity clusters. The paper is logical and informative. It is a novel and interesting topic. I suggest it to be accepted after addressing the following comments.

**Response:** We appreciate very much your encouraging comments. In the followings, we have carefully considered and responded to your specific comments and suggestions.

1. L91-93, The increase in the contribution of $O_3$-exceeding days is the primary contributor to the large increase in the annual mean $O_3$. The contribution of $O_3$-exceeding days is affected by exceeding days and concentrations. Therefore, I think the mean concentration of $O_3$-exceeding days should be shown.

**Response:**

Thank you for an excellent suggestion. We have revised L91–93 in the revised manuscript to: "It is clear that the increase in $O_3$-exceeding days is the primary

contributor to the large increase in the annual mean $O_3$ in all three megacity clusters from 2015 to 2020. The contribution of $O_3$-exceeding days is affected mostly by the changing number of exceeding days (more than 80%), and secondly but still significantly by their changes in concentrations (less than 20%) (Tables 2–4). E.g. in BTH the exceeding days were 31, 43, 62, 74, 96 and 78 days in the individual years of 2015–2020, respectively, while their concentrations of those years were 66.42, 64.13, 69.44, 68.21, 70.19 and 69.69, respectively (Table 2 second column). Contributions from non-$O_3$-exceeding days are insignificant ($p > 0.1$), except that in BTH (Figure 2a) which shows a significant declining contribution ($p = 0.02$) due to the reduced number of non-$O_3$-exceeding days."

2. L162-163, The contribution of the increased $O_3$ concentration and the number of $O_3$-exceeding days is +20% and +34%, respectively. Please elaborate more about this method, how is the contribution calculated? L214-219, and the difference in $O_3$ between 2017 and 2015 can be attributed to the large number of days and higher average concentration with four or more consecutive $O_3$-exceeding days and those with less than four consecutive $O_3$-exceeding days. How are the respective contributions distinguished?

**Response:**

Line 162–163: Accounted for the number of $O_3$-exceeding days, the ratio of MDA8 $O_3$ in all $O_3$-exceeding days between 2017 and 2015 became $(64.35 \times 40)/(53.79 \times 31)=1.54$. The combined effect of ozone concentration and $O_3$-exceeding days was 54%, with the effect of ozone concentration is $(64.35-53.79)/53.79=0.2(20\%)$, so the effect of $O_3$-exceeding days is 54%-20%=34%. We acknowledge that the above calculation method shortchanges slightly the contribution of the increased $O_3$ concentration compared to the calculation in which the contribution of the number of $O_3$-exceeding days is calculated first, i.e. $(40-31)/31=29\%$ is the contribution of the number of days, and 54%-29%=25% the contribution of the $O_3$ concentration. This shortchanging effect is caused by the difficulty in attributing the cross term between the number of days and

the $O_3$ concentration.

Line 214–219: The respective contributions were distinguished by first evaluating the contribution due to Line 214–216 "One of the most remarkable differences between 2017 and 2015 in Figures 9a–9f was the large number of days with four or more consecutive $O_3$-exceeding days in 2017 (28 days, Figure 9d) over that of 2015 (7 days, Figure 9a)": 28-7=21, the difference in total $O_3$-exceeding between 2017 and 2015 (Table 2, column 2) is 62-31=31, 21/31=68% which is fairly close to the number 62% in Line 216. Secondly, the contribution of Line 217–218 "the 10 days' difference (2017 vs. 2015) in the number of days with less than four consecutive $O_3$-exceeding days" is 10/31=32%, which is fairly close to the number 30% in Line 217. The discussion above explains how the major respective contributions were distinguished. The small discrepancies of a few percent is due to the changes in $O_3$ concentrations.

3. L181-184, The emission of air pollutants and $O_3$ precursors isn't a main cause of the expansion and saturation in BTH. Is the result the same in PRD and YRD?

**Response:**

Yes, the results in PRD and YRD are same as BTH, as illustrated below:

[Figure]

Figure R1. Annual mean concentrations of maximum daily 8-hour average $O_3$ in YRD during $O_3$-exceeding days for all stations (black), polluted stations (red) and clean stations (green) (a), same as (a) except for (b) Ox, (c) $NO_2$, (d) CO, (e) $PM_{2.5}$ and (f) $SO_2$.

[Figure]

Figure R2. Same as Figure R1, except for PRD.

4. The values of SSR and T2m of the $O_3$ episodes with four or more consecutive $O_3$-exceeding days is lower than those of $O_3$ episodes with less than four consecutive $O_3$-exceeding days in PRD during 2017-2020, which are different with in BTH and YRD. In addition, $O_3$-exceeding days in PRD are mostly decoupled from those in BTH and YRD. Does it imply that the cause of worsening $O_3$ trend in PRD is different with BTH and YRD?

**Response:**

This is a very good and challenging question. Yes, we believe that specific causes of worsening $O_3$ trend are different among PRD, BTH and YRD. Nevertheless, the causes are most likely meteorological/climate oscillation in nature. For instance, we think that changes in tropical cyclones might be a factor in the worsening $O_3$ trend in PRD and southern YRD.

5. L280-284, The annual weighted SSR/T2m can be performed by multiplying the difference between the four or more consecutive $O_3$-exceeding days and clean days of SSR/T2m with the frequency of occurrence of $O_3$ episodes with four or more consecutive $O_3$-exceeding days each year. Is that right? Has this approach been used in previous studies? Do the months inside the parentheses in Figure 12 represent the months when O3-exceeding days happen in this region?

**Response:**

L280–284, Yes, you are right about the approach used to get the results in Figure 12. To our limited knowledge, this approach has not been used in previous studies.

Yes, the months in parentheses in Figure 12 represent the months in which the area experienced four or more consecutive $O_3$-exceeding days.

6. Why did the expansion and saturation occur mostly during 2015-2017? $O_3$ concentrations have increased in 2015-2019. The annual mean concentration of MAD8 decreased significantly in 2020. In BTH, YRD and PRD, the number of days of all $O_3$-

exceeding days has increased from 2015 to 2019. However, the number of O₃-exceeding days decreases in 2020. Is this change also due to the influence of WPSH and mid-high latitude wave activities?

**Response:**

The statement of "the expansion and saturation occur mostly during 2015–2017" was entirely based on the results shown in Figure 8: the green line and the red line converge to each other from 2015 to 2017.

Our view is that meteorology and climate played the major role in the interannual variations and trends in ozone in the three megacity clusters during 2015–2020, but it does not necessarily apply to the specific problem of "the number of O₃-exceeding days decreases in 2020".

7. Figure 7 is the same as Figure 6. Please check it.

**Response:**

Sorry! In the version of this article initially submitted, Figure 7 was erroneously pasted. The correct Figure 7 is shown in the following:

[Figure]

Figure R3. Spatial distribution of annual mean concentrations of maximum daily 8-hour average $O_3$ for $O_3$-exceeding days in PRD in 2015 (a), 2017 (b) and their difference (2017–2015) (c). The top, middle and bottom rectangle boxes denote BTH, YRD and PRD districts, respectively. The number inside the parenthesis behind 2015 or 2017 denotes the number of $O_3$-exceeding days.

---

## Author Comment (AC3)

Dear Editor,

We appreciate the prompt reviews and would like to thank the reviewers for insightful comments and suggestions on our manuscript entitled "What is the cause(s) of ozone trends in three megacity clusters in eastern China during 2015–2020?" (MS No.: acp-2022-781). We have carefully considered all comments and suggestions. Listed below are our point-by-point responses to all comments and suggestions of Reviewer #2 (Reviewer's points in black, our responses in blue).

**Anonymous Referee #2**

This paper analyzes the causes of the 2015-2020 surface ozone increases over three megacity clusters in China (BTH, YRD, PRD) and concludes that increasing Western Pacific Subtropical High (WSPH) conditions are responsible for the ozone increase rather than changes in emissions.

I found the paper difficult to read because it is so chatty, hand-waving, and qualitative and weak in its argumentation. Its central thesis that the ozone trend is driven by the WSPH is in my opinion unsupported and flies in the face of ample literature showing that ozone trends in China over the past decade are anthropogenically driven including as evidence (1) removal of meteorological influence using statistical models, (2) broadening of the ozone season, (3) surge of ozone following the Covid shutdown, (4) consistency with model ozone increases when using anthropogenic emission trends as input. The paper largely ignores this literature, but if it is to make the contrary claim that the ozone trend is in fact driven by meteorology rather than emissions it either needs to refute or show consistency with these different strands of evidence. It does not.

The proposed evidence for a WSPH driver of ozone trends is in my opinion very flimsy. The first piece of evidence proposed is that ozone pollution episodes are becoming more regional, but (a) this does not imply a meteorological trend (witness the US in the 1980s when the same phenomenon was observed), (b) it could reflect

the well-known broadening of the ozone season in China (a big weakness of this paper is not resolving the seasonal variation of ozone). The second piece of evidence proposed is the correlation of ozone with temperature and SSR, combined with the trends in these meteorological variables over 2015-2020, but (a) this correlation with meteorological variables is well known, (b) past studies have removed statistically the influence of meteorology on the ozone trend, as is very standard practice.

So I don't think that this paper should be published in anything close to current form. It could be used to argue wrongly against the urgency for China to decrease VOC emissions, and it will waste the research community's time in having to debunk it. One interesting result in this paper is the apparent regionalization of ozone pollution in China, which I don't think has been discussed before. That could provide the basis for a paper but it would need to be better demonstrated.

**Response:**

With due respect, we are surprised that this referee made extensive critical comments from the point of view of a believer in photochemical/emission cause of the ozone trend, rather than an objective referee. This referee's subjective critical comments are clearly demonstrated in the concluding remarks quoted below: "So I don't think that this paper should be published in anything close to current form. It could be used to argue wrongly against the urgency for China to decrease VOC emissions, and it will waste the research community's time in having to debunk it." In our opinion, publication of a paper should be judged by its scientific merit, not by its possible impact on the environmental policy in China as suggested by the referee. Besides, the meteorology cause of the ozone trend could very well be argued as a reason to call for a more stringent control of VOC emissions. Our responses to other tangible criticisms are presented in the following.

(a) Our presentations are "hand-waving": We presented 12 figures and 4 tables in the manuscript, each and every one of them was based on observed data available to

the public.

(b) "it could reflect the well-known broadening of the ozone season in China (a big weakness of this paper is not resolving the seasonal variation of ozone)" According to our analysis, the variation in consecutive $O_3$-exceeding days was the key driver for the trend in the annual $O_3$ concentration (Figure 2). So we made most of our analysis for the period consecutive $O_3$-exceeding days occurred. E.g. April–September in BTH. We found no broadening of the consecutive $O_3$-exceeding days beyond this period. Within this period, the broadening of the consecutive $O_3$-exceeding days was addressed implicitly in our analysis.

(c) "past studies have removed statistically the influence of meteorology on the ozone trend, as is very standard practice" We believe removing statistically the influence of meteorology on the ozone trend is a useful, but not a deterministic method for the ozone trend analysis, because current state of knowledge of meteorology-ozone relationship is inadequate for determining which meteorological parameters or mechanisms to remove.

**Specific comments:**

1. Line 65: are only sites with complete records for 2015-2020 used? Otherwise the analysis would be biased by expansion of the network.

**Response:**

For each site, the maximum daily 8-h average concentration (MDA8) of $O_3$ is calculated by utilizing an 8-h moving average window for each day. To ensure the data quality, the 8-h moving window has to follow Technical Regulation for Ambient Air Quality Assessment (on trial) issued by the Ministry of Environmental Protection of China. Technical regulation states that the maximum 8-hour average validity of $O_3$ within a natural day is defined as at least 14 effective 8-hour average concentrations from 8:00 to 24:00 on that day. When 14 valid data are not satisfied, the statistical

results are still valid if the daily maximum 8-h average concentration exceeds the concentration limit. In our study, 78 among the 80 sites in BTH, 152 among the 181 sites in YRD, 48 among the 56 sites in PRD, are selected. Therefore, no significant bias is expected.

2. Line 83: I didn't see the criteria for separating clean and polluted sites in Table 1.

**Response:**

Thank you for pointing out this oversight in the manuscript. Clean sites are defined as MDA8 $O_3$ concentration less than 73 ppb in 2015. Polluted sites are defined as MDA8 $O_3$ concentration more than 92 ppb in 2015.

3. Line 125: text and captions don't match what is actually shown in Figures 5-7. Comparing just two years of data (2017 vs. 2015) as a trend indicator is obviously bad – of course the difference between two individual years could be meteorologically driven.

**Response:**

In the version of this article initially published, Figure 7 was erroneously pasted. Please see the errata we posted on January 6, 2023 at the public discussion site link.

In regard to "Comparing just two years of data (2017 vs. 2015) as a trend indicator is obviously bad", we agree with this comment. But the trend we are dealing with is only six years (2015–2020) which normally would not be referred to as a trend if not because of its important environmental impacts. In addition, thank you for agreeing on "of course the difference between two individual years could be meteorologically driven". These two years represent one third of the entire period and contribute more than one third of the entire trend (Figure 1).

4. Line 136, elsewhere: a big weakness of this paper is not resolving the ozone data by season as is standard practice. In particular, it is not clear to me that this

regionalization of ozone could not simply reflect the broadening of the ozone season that has been reported in previous papers.

**Response:**

We agree with the reviewer's opinion that ozone pollution is obviously seasonal and regional. We have already addressed the issue of broadening of the ozone season in the response to the general comments as follows:

According to our analysis, the variation in consecutive $O_3$-exceeding days was the key driver for the trend in the annual $O_3$ concentration (Figure 2). So we made most of our analysis for the period consecutive $O_3$-exceeding days occurred. E.g. April–September in BTH. We found no broadening of the consecutive $O_3$-exceeding days beyond this period. Within this period, the broadening of the consecutive $O_3$-exceeding days was addressed implicitly in our analysis.

5. Line 141: why jump to an attribution to weather? This is characteristic of the weak argumentation throughout this paper. Same thing in line 151 – why would the regionalization of BTH and YRD be very unlikely to be driven by emissions?

**Response:**

From Figure 8 in the manuscript and Figures R1 and R2, it can be seen that $NO_2$ and CO did not change significantly between 2015 and 2017. In addition, it can be seen in Figure R3 that the emission of NOx and VOCs in the three regions remained basically unchanged. Hence we conclude that the emission of air pollutants and $O_3$ precursors isn't a major cause of the expansion and saturation in BTH, YRD or PRD.

[Figure]

**Figure R1.** Annual mean concentrations of maximum daily 8-hour average $O_3$ in YRD during O3-exceeding days for all stations (black), polluted stations (red) and clean stations (green) (a), same as (a) except for Ox (b), $NO_2$ (c), CO (d), $PM_{2.5}$ (e), $SO_2$ (f).

[Figure]

**Figure R2.** The same as Figure R1 but for PRD.

[Figure]

**Figure R3.** NOx and VOCs emissions in the BTH, YRD and PRD regions form 2015 to 2020.

6. Line 184: no one is arguing that the increase in ozone is driven by decreasing NO titration (OK, maybe in winter, but ozone is low then anyhow). The argument is that it is driven by NOx emission decreases under VOC-limited conditions.

**Response:**

Li et al.(2022) convincingly demonstrated that the small yet significant increasing ozone trend in PRD in 2006–2018 was caused by the decrease of the titration effect of NO. However, our analysis of the recent large positive ozone trend in PRD in 2015–2020 showed that the trend was not driven by decreasing NO.

**References:**

Li, X., Yuan, B., Parrish, D. D., Chen, D., Song, Y., Yang, S., Liu, Z., Shao, M.: Long-term trend of ozone in southern China reveals future mitigation strategy for air pollution, Atmos. Environ., 269, 118869, https://doi.org/10.1016/j.atmosenv.2021.118869, 2022.

7. Line 187: past studies have attributed the ozone increase to $PM_{2.5}$ decrease only for summer. Again, the paper would need to resolve its analysis by season.

**Response:**

We follow this suggestion to make an analysis for summer data, and add the results as Figures R4–R5 for comparison. we found the results of summer season were similar

to those of April–September, which was the time period studied in the original manuscript.

[Figure]

**Figure R4.** The same as Figure R1 but for BTH in April–September.

[Figure]

**Figure R5.** The same as Figure R1, but for BTH in June–August.

8. Line 241, elsewhere: there is nothing new in the attribution of ozone pollution episodes to the WSPH. The lengthy discussion of this attribution is just routine.

**Response:**

We agree that "there is nothing new in the attribution of ozone pollution episodes to the WSPH". We never pretended it was new as evident by many citations of previous studies, e.g. Line 35–37: "In addition, large-scale circulations, such as the East Asian monsoon, West Pacific subtropical high (WPSH) and tropical cyclones can influence $O_3$ concentration as well (Yang et al., 2014; Zhao and Wang, 2017; Lu et al., 2019; Rowlinson et al., 2019)", and Line 51–54 "we noticed that the interannual variations of $O_3$ concentration were strongly affected by the position and intensity of WPSH and the presence tropical cyclones in the western Pacific and South China Sea, consistent with the results of a number of recent studies (Zhao and Wang, 2017; Chang et al., 2019; Yin et al., 2019; Mao et al., 2020; Ouyang et al., 2022)."